# Resilience among Turkish adolescents: A multi-level approach

**Gokhan Cakir[1], Utku Isik[2]\*, Umit Dogan Ustun[1], Nihan Su[3], Osman Gumusgul[4]**

**1** Faculty of Sports Science, Recep Tayyip Erdogan University, Rize, Türkiye, **2** Faculty of Sports Science, Ege University, İzmir, Türkiye, **3** Faculty of Sports Science, Kocaeli University, Kocaeli, Türkiye, **4** Faculty of Sports Science, Kutahya Dumlupinar University, Kütahya, Türkiye

\* utku.isik@ege.edu.tr

## Abstract

The objective of this study is to determine the components that contribute to psychological resilience in adolescents and to determine if physical exercise, emotion control, or self-efficacy are more effective predictors of resilience. Data from participants was collected through a personal information form, the International Physical Activity Questionnaire— Short Form, the Self-Efficacy Scale for Children, the Emotion Regulation Scale for Children and Adolescents, and the Psychological Resilience Scale for Children and Adolescents. The data were gathered online from 16 out of the 81 provinces in Turkey, representing 7 different regions, using convenience sampling. The study sample comprised 505 adolescents, with 309 females and 196 males. The average age of the participants was 15.66 years, with a standard deviation of 1.34. The data obtained from the students was analyzed using SPSS 27.0 statistical software. The Chi-Square test was employed to establish the correlation between the demographic features of adolescents and their levels of physical activity. The relationship between the independent variables and the dependent variable was determined using correlation analysis and hierarchical regression analysis. According to the analyzed results, there was a substantial positive correlation between physical activity and resilience, reappraisal and resilience, and self-efficacy and resilience. In addition, it was noted that physical activity alone explains 4.8% of the overall variation in resilience and is a significant predictor of resilience. The inclusion of reappraisal in the model resulted in a partial prediction of resilience by physical activity. However, the primary strength of the model was attributed to reappraisal. The inclusion of self-efficacy in the model resulted in a significant prediction of resilience, accounting for 36.8% of the total variance. The self-efficacy variable had a higher impact level compared to the other variables. Furthermore, the inclusion of self-efficacy in the model resulted in the elimination of the influence of physical activity on resilience. The research conclusions point out that self-efficacy has a greater impact on psychological resilience compared to physical activity and emotion regulation.

## Introduction

The mental well-being of adults is significantly influenced by their mental condition during childhood and adolescence [1]. Adolescence is commonly known as the "danger zone,"

**Data Availability Statement:** All data files are available from the Figshare. 10.6084/m9.figshare. 25706454 https://figshare.com/s/ 7f847cdab0637d9ee662.

**Funding:** The author(s) received no specific funding for this work.

**Competing interests:** The authors have declared that no competing interests exist.

according to the World Health Organization [2], as cognitive, emotional, and behavioral growth may be affected as a result of difficult emotional reactions such as separation from parents, identity formation, and individuation [3,4]. The mental well-being of teenagers is intricately connected to their social growth [5]. Today's perpetually changing and evolving social environment has placed adolescents under unprecedented levels of stress [6]. The presence of feelings of rejection and judgment in social situations can diminish an individual's confidence, potentially resulting in the development of more enduring issues in the future. During adolescence, individuals are susceptible to a range of mental and physical developmental vulnerabilities and dysfunctions, including mood disorders, substance misuse, and obesity [7,8]. Hence, it is crucial to ascertain strategies for effectively managing the diverse problems and pathologies encountered by teenagers, with the aim of fostering a healthier transition into adulthood [9]. Psychological resilience is a highly effective strategy for dealing with the difficulties of adolescence. Psychological resilience is positively and significantly associated with levels of mental health [10,11]. The study focuses on the dependent variable of psychological resilience, which is defined as the capacity to effectively adjust to and cope with life's challenges and adversities [12]. Resilience is a dynamic developmental process that promotes positive adaptation to stressful, adverse and traumatic circumstances [13]. In other words, resilience has been conceptualised as trait resilience, which is "a characteristic that moderates the negative effects of stress and promotes adaptation" [14].

Resilient individuals have the ability to ascribe favorable interpretations to intricate occurrences, manage adverse emotions, and adjust to evolving external stressors over the course of their lives [15]. Insufficient levels of psychological resilience in teenagers can cause a delay in their mental and social development, as well as a lack of independence and responsibility. This can then contribute to the development of gaming addiction, excessive anxiety, academic pressure, and other related symptoms [16]). Additionally, it can result in melancholy, an incapacity to manage stress, suicide, subpar academic achievement, and other troublesome behaviors [17]. Recent studies have observed a concerning rise in depression and anxiety among teenagers and young adults, leading to decreased levels of life satisfaction. These issues have notably intensified following the onset of the pandemic [18,19]. Studies examining this situation have shown that resilience and hope are effective factors in predicting stress arising from the Pandemic [20]. At this point, it is crucial to develop resilience, as it is a predictor of stress and determinant of the life satisfaction of adolescents. However, can resilience be developed? Selçuk [4] argues that the psychological resilience of individuals is a dynamic construct, therefore necessitating an understanding of resilience as an evolving process. In other words, there are concepts that feed and develop the resilience of individuals in a wide range from childhood to adulthood, such as art, play, physical activity, self-efficacy, mindfulness, meditation, self-regulation, and emotion regulation. So, the present study aims to determine which of physical activity, emotion control, and self-efficacy is a more precise indicator of psychological resilience when these concepts are seen as determinants of individual resilience. Providing an explanation of the conceptual framework is beneficial in understanding the rationale behind the concentration on these three factors.

## Theoretical background

**Physical activity and resilience.** Physical activity has a positive impact on the social adjustment development of adolescents [21]. In addition, physical activity has been found to provide protection against mental health disorders such as depression and tension [22,23]. Evidence showing the beneficial impact of physical activity on mental health has consistently persisted in the literature throughout modern science. A study conducted with a sample of

university students revealed that engaging in physical activity had a positive impact on psychological resilience [24].

While all forms of physical activity are recognized to be beneficial for overall well-being, individuals who participate in high-intensity exercise demonstrate greater psychological resilience, as observed by Szuhany et al. [23]. Current studies indicate that engagement in physical activity during adolescence might reduce unpleasant emotions. Engaging in regular physical activity can have a transformative effect on mental well-being, converting feelings of worry, despair, and stress into good emotions like enjoyment, joy, and relaxation [25,26]. To clarify, encouraging physical activities that boost psychological resilience may serve as a means to increase the mental health of adolescents [27]. Consistent with these studies, it is anticipated that the physical activity levels of adolescents may impact their resilience. Given the specifics, the subsequent hypothesis was formulated:

H1: There is a relationship between physical activity and adolescent resilience.

**Emotional regulation and self-efficacy as factors influencing psychological resilience.**
Emotional regulation is a crucial notion that promotes psychological resilience. Emotional regulation is a technique that aims to modify the influence of emotions on people [28]. The emotional regulation approach, as identified by Gratz and Roemer [29], is one of the frameworks that helps in managing emotions. Regulating one's emotions in the face of challenging situations is seen as a crucial procedure for fostering optimal psychological functioning in adolescents [30]. The ability to regulate emotions might be seen as a valuable tool that protects adolescents from engaging in unreasonable and harmful behaviors [31]. As cited in Nooripour et al. [32] researchers believe that inadequate regulation of emotional responses may be a significant factor in developing aggression, feelings of guilt through incitement to violence, and shame. So, adolescents must prioritize the development and utilization of positive self-regulation skills in order to strengthen their psychological resilience [33]. Reappraisal and suppression are two separate strategies highlighted in the literature on emotion regulation. Reappraisal is conceptualized as a cognitive change strategy, while suppression is considered a response modulation strategy. But, due to its structure, the suppression strategy can cause individuals to behave inauthentically and alienating themselves. It can also prevent them from developing close relationships and make them feel uneasy, nervous, and shy in interpersonal situations. Besides, suppression strategies have been found to have adverse effects, leading to depressive symptoms. On the other hand, reappraisal strategies show a positive connection with these adverse effects. Consequently, reappraisal strategies are thought to be linked to resilience [28].

Self-efficacy is another concept that contributes to the development of psychological resilience [34]. Self-efficacy refers to an individual's belief in their ability to perform well in a specific environment and achieve goals related to a task [35]. Researchers have conceptualized self-efficacy as an expression of one's general confidence in dealing with effort-demanding or novel situations [36]. At this juncture, a relationship between self-efficacy and resilience is postulated. For example, in a group of 139 Italian adolescents Sagone and De Caroli [37] proved the existence of significant relationships between resilience and both generalized and scholastic self-efficacy. In an another study conducted with 302 late adolescents Develos Sacdalan and Bozkuş [38] found resilience as a full mediator in the relationship between self-determination and self-efficacy.

Self-efficacy helps one to develop motivation and envision challenging goals in life [39]. The correlation between psychological resilience and self-efficacy in adolescence is widely regarded as an essential variable in promoting positive adaptation throughout this phase of

growth. The link under examination assesses the impact of perceived self-efficacy on life skills, specifically in terms of overcoming problems [40]. Given the circumstances, the subsequent hypotheses have been formulated:

H2: Emotion regulation positively affects adolescents' resilience through reappraisal.

H3: Self-efficacy positively affects adolescent resilience.

**Present study.** Psychological interventions targeting adolescents' resilience can reduce stress, improve their overall well-being and mental health, and cultivate their long-term coping abilities [41,42]. Adolescence is a stage in life where young people are susceptible to mental and physical vulnerabilities and limitations in functioning. Thus, it is essential to cultivate elements that foster psychological resilience in order to maintain the overall well-being of adolescents, both physically and mentally. The primary objective of this study is to ascertain the determinants that contribute to the enhancement of psychological resilience among adolescents. Potential predictors that significantly influence resilience include physical activity, emotional regulation, and self-efficacy. Gaining insight into the impact of these elements on the psychological resilience of adolescents is crucial for enhancing their ability to manage stress and promote a sound mental well-being. Besides, after conducting a thorough examination of the existing literature only a few studies have been found that clarify and assess the correlation between physical activity and resilience by employing multiple variables. Expanding upon the concept that physical, mental, and emotional aspects may be interrelated, it is believed that addressing factors that can improve adolescents' resilience in a comprehensive manner might address certain deficiencies in the existing literature. The findings of our study have the potential to contribute to the formulation of effective approaches aimed at enhancing the psychological resilience of adolescents.

## Materials and methods

### Research model

This research utilized the correlational survey model, a quantitative research method. According to Büyüköztürk [43], correlational survey designs determine the presence and/or degree of change between two or more variables. This research examined physical activity, self-efficacy, and emotion regulation as independent variables. The dependent variable was resilience. Thus, this research model sought to identify psychological resilience determinants and determine which physical activity, emotion regulation, and self-efficacy better predict resilience (Fig 1).

### Participants

Cross-sectional convenience sampling was used in this study. During the research, data was collected from 621 participants. 18 individuals (2.8%) under the age of 14 and over 19, as well as 55 participants (9.1%) with incomplete or randomly filled questionnaires, were excluded from the analysis. Moreover, to identify outliers, the %5 trimmed value in the descriptive table was checked, resulting in the exclusion of 43 data points (7.8%) from the analysis. Consequently, data from 505 adolescent participants sampled conveniently were evaluated within the scope of the analysis.

Of the 505 participants, 196 (38.8%) were male, and 309 (61.2%) were female, with ages ranging from 14 to 19 years (15.65 ± 1.34). The number of participants engaged in sports and those not engaged in sports were 135 (26.7%) and 370 (73.3%), respectively. According to the

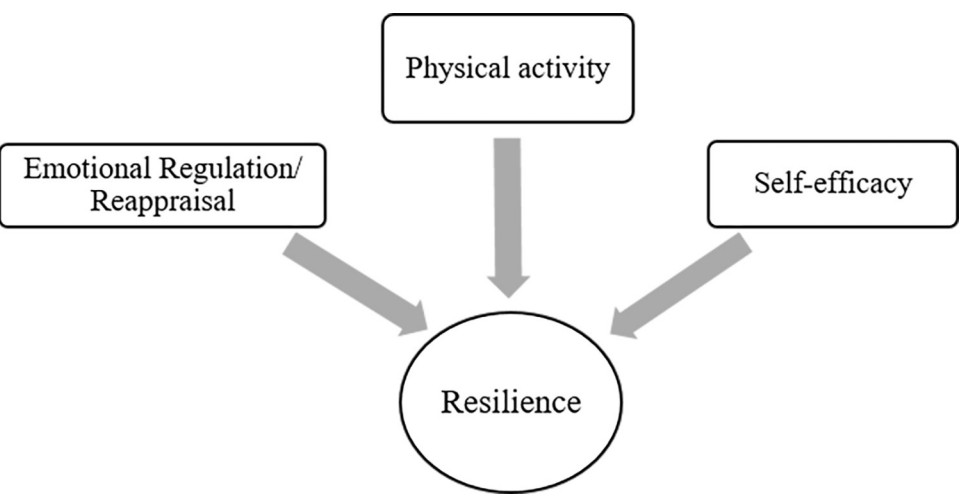

**Fig 1. Research model.**

International Physical Activity Questionnaire, 4.1% of the participants were found to have high physical activity levels. Table 1 displays the numerical distribution of participants based on their physical activity levels. Significant differences in physical activity levels based on sports engagement and gender were identified through chi-square test analyses (p<0.05). Participants with generally low and moderate physical activity levels were found to have lower participation in sports, while those with high physical activity levels had a higher rate of sports participation. Additionally, there were more male participants with low and moderate physical activity levels, whereas there were more female participants with high physical activity levels (p>0.05).

## Procedure, and ethics

All data were collected between October 1, 2023, and November 30, 2023. Prior to administering the surveys, verbal consents and permission letters were obtained from the school principals. The obtained permission letters than were submitted to the University ethics committee. Ethics approval was given by the . . . . . . . . . . . . University Ethics Committee on 27.09.2023 with the decision number 2023/245. The inclusion criteria for the study are as follows:

**Table 1. Frequencies according to participants' physical activity level.**

| | | All (%) | Low (%) | Mid (%) | High (%) | p-value |
|---|---|---|---|---|---|---|
| Class | | | | | | |
| | Grade-1 | 25.545 | 23.232 | 26.744 | 25.000 | 0.287 |
| | Grade-2 | 16.832 | 10.101 | 19.767 | 16.216 | |
| | Grade-3 | 18.416 | 20.202 | 16.279 | 20.946 | |
| | Grade-4 | 39.208 | 46.465 | 37.209 | 37.838 | |
| Engaged in Sports | | | | | | |
| | Yes | 26.733 | 11.111 | 14.729 | 58.108 | p<0.001 |
| | No | 73.267 | 88.889 | 85.271 | 41.892 | |
| Gender | | | | | | |
| | Men | 61.188 | 84.848 | 65.116 | 38.514 | p<0.001 |
| | Women | 38.812 | 15.152 | 34.884 | 61.486 | |

Adolescents who aged 14–19 voluntarily agree to participate in the study. The exclusion criteria of the study are as follows: Adolescents who fill out survey and/or data collection forms incompletely. The data were collected by the participants themselves on predetermined dates. Participation in the study was voluntary, and in addition to verbal consent an informed consent form was obtained from the parents of participants under the age of 18. Prior to completing the surveys, participants were provided with essential explanations. During the data collection process, students were instructed to respond to the scales in the most suitable manner, thereby indicating the absence of any definitive correct or incorrect answer. Participants required approximately 15 minutes to complete the surveys.

## Data collection tools

In this study, the researchers utilized the Children's Self Efficacy Scale, Child and Adolescent Emotion Regulation Scale, Child and Adolescent Psychological Resilience Scale (CAPS-12), and International Physical Activity Questionnaire (Short Form) as assessment tools deemed appropriate for the research objectives.

## Personal information form

The researcher created this form to gather demographic data about the study participants' gender, age, and level of physical activity.

## Children's self efficacy scale

Originally developed by Muris [44], and the Turkish adaptation study was conducted by Telef and Karaca [36]. The scale assesses the self-efficacy of adolescents across three dimensions: social, academic, and emotional. Additionally, it evaluates the general self-efficacy by summing the scores derived from all items. The scale was employed in the study due to its established validity and reliability specifically on adolescents, as well as its ability to assess general self-efficacy. The Children's Self-Efficacy Scale is a five-point Likert-type scale (1 = not at all and 5 = very well). The highest score that can be obtained from the scale is 105, and the lowest score is 21. A high score on the scale indicates a high level of general self-efficacy related to participant, and a low score on the scale indicates a low level of self-efficacy. When the internal consistency coefficients of the Children's Self-Efficacy Scale were examined, a coefficient of .86 was calculated for the scale as a whole. In this study, the internal consistency coefficient was calculated as .87.

## Child and adolescent emotion regulation scale

The original scale was developed by Gullone and Taffe [45], and the Turkish adaptation study was conducted by Tetik and Önder [28]. The scale consists of two separate dimensions, defined as reappraisal and suppression. Reappraisal is a cognitive change strategy. It is defined as a reinterpretation attempt to change the meaning of the experienced situation and is associated with resilience. Therefore, this subscale of the scale was used within the scope of the study. The Cronbach alpha internal consistency coefficients of the sub-dimensions are .79 for the reappraisal sub-scale [28]. In this study, the internal consistency coefficient was calculated as .71 for the reappraisal sub-scale.

## Child and adolescent psychological resilience scale

The scale was initially developed by Liebenberg et al. [46] and subsequently adapted in Turkish by Arslan [47]. The scale consists of a total of 12 items and is structured as a single factor. In

the literature, various measurement tools have been employed to assess psychological resilience. However, the present study opted to utilize the current scale due to its single-factor structure, which enables an evaluation to be conducted based on the overall score. Within the scope of the reliability study, the internal consistency coefficient was calculated for the scale as .91. In this study, the internal consistency coefficient was calculated as .80.

### International physical activity questionnaire (Short form)

The validity and reliability of the International Physical Activity Questionnaire Short Form in Turkish was conducted by Savcı et al. [48]. This self-report scale assesses individuals' walking habits, engagement in moderate and vigorous activities, and duration of sitting for the past seven days. The total score of the short form is determined by adding up the duration (measured in minutes) and frequency (measured in days) of walking, moderate activity, and vigorous activity. The calculation of energy expenditure for various activities is determined using the MET-minute score. These activities have been assigned standard MET values. Furthermore, classification is conducted based on the numerical data acquired, in addition to the ongoing scoring process. The classification consists of three categories: (1) Inactive type, (2) Minimal active type, and (3) Very active type. Given its widespread usage and international validity, we employed this scale in our study.

### Data analysis

In the data analysis, G*Power was used to determine the sample size in the data analysis (3.1.9.7) [49]. The calculation indicated that a total of 439 participants would be sufficient. This determination was made with a power of $(1-\beta) = 0.95$, a small effect size $(f^2) = 0.04$, and $\alpha = 0.05$, considering three predictor variables [26,50]. For hierarchical regression and other multivariate analyses, a sample size of 40 times the independent variables were also found to be sufficient [51]. Descriptive statistics were used to reveal the demographic characteristics of the participants. In this stage, the mean and standard deviation values of all variables were calculated, and the normality assumption was checked by examining the skewness and kurtosis values. The ±2 base of George and Mallery [52] was taken as the reference value for the normality assumption.

The relationship between adolescents' demographic characteristics and their physical activity levels was determined using a chi-square analysis. The impact of independent variables on the dependent variable was established through hierarchical regression analysis. The main reason for using hierarchical regression analysis was to clearly reveal which variable predicts resilience better. Prior to conducting these analyses, assumptions were examined. For each model, regression analysis assumptions including normality, linearity, autocorrelation (Durbin-Watson test value = 2.70), constant variance (homoscedasticity), detection of outliers (DFBeta, Cook's Distance), and multicollinearity were tested [51], and it was observed that the assumptions were met. The collinearity diagnostics showed that the VIFs of the study variables were below the recommended threshold of 5, demonstrating a low concern for multicollinearity in the data [53]. Histogram graphs, skewness and kurtosis values, Levene test and Box *M* test values of dependent variables were also examined for univariate normality. Additionally, a correlation analysis was performed before the hierarchical regression analysis to determine the relationship between the variables.

## Results

### Descriptive statistics and correlation analysis

Table 2 presents the means, standard deviations (SD), and Pearson's correlation coefficients for Physical Activity, reappraisal, suppression, resilience, and self-efficacy. The results

**Table 2. Descriptive statistics and Pearson correlations of the study variables.**

|  | Mean | SD | 1 | 2 | 3 | 4 | 5 | 6 | 7 |
|---|---|---|---|---|---|---|---|---|---|
| 1.Gender |  |  | 1 |  |  |  |  |  |  |
| 2.Age | 15.650 | 1.340 | 0.057 | 1 |  |  |  |  |  |
| 3.Engaged in Sports |  |  | -.318** | -0.019 | 1 |  |  |  |  |
| 4.Physical Activity | 2681.09 | 3194.11 | .307** | 0.037 | -.461** | 1 |  |  |  |
| 5.Reappraisal | 20.74 | 3.34 | .125** | -0.014 | -.162** | .214** | 1 |  |  |
| 6.Resilience | 45.51 | 6.84 | .125** | -.095* | -.209** | .220** | .411** | 1 |  |
| 7.Self-efficacy | 66.38 | 13.25 | .247** | -0.024 | -.138** | .251** | .463** | .582** | 1 |

Note: M = mean; SD = standard deviation. Gender is a dummy code (female = 1; male = 2). Engaged in Sports is a dummy code (yes = 1; no = 2).

*p < 0.05

**p < 0.01.

demonstrated a significant correlation between the study variables. Physical activity was significantly positively correlated with reappraisal (r = 0.214, p < 0.001), resilience (r = 0.220, p < 0.001), and self-efficacy (r = 0.251, p < 0.001). Reappraisal was significantly positively correlated with suppression (r = 0.089, p < 0.05), resilience (r = 0.411, p < 0.001), and self-efficacy (r = 0.463, p < 0.001). Resilience was significantly and positively correlated with self-efficacy (r = 0.582, p < 0.001). The significant correlations among the study variables provide initial support for the proposed hypotheses.

The hierarchical regression analysis used to find the variables that could predict the resilience of the participants showed that physical activity fit into the first model, reappraisal from the emotion regulation sub-dimensions fit into the second model, and self-efficacy fit into the third model (Table 3).

Based on the analysis presented in Table 3, it was found that in all three models, participants significantly predicted resilience values. Upon examining Model 1, it was observed that physical activity alone accounts for 4.8% of the total variance of resilience ($R = 0.220$; $R^2 = 0.048$; $F_{(1,503)} = 25.495$; $p = 0.000$; Effect size($f^2$) = 0.05) and significantly predicts resilience. In Model 2, when reappraisal, a sub-dimension of emotion regulation, was included in the model, physical activity still predicted resilience to a certain extent ($\beta = 0.142$, p<0.05), but the main strength of the model was provided by reappraisal ($\beta = 0.381$, p<0.01). In other words, when

**Table 3. Analysis of hierarchical regression between variables.**

| Dependent | Predictors | B | Std. Error | Beta | t | p | %95 CI |
|---|---|---|---|---|---|---|---|
| Resilience (Model 1) | (Constant) | 44.259 | 0.388 |  | 114.068 | 0.000 | [43.496, 45.021] |
|  | Physical Activity | 0.000 | 0.000 | 0.220 | 5.049 | 0.000 | [.00, .01] |
| $R = .220$; $R^2 = .048$; $F_{(1,503)} = 25.495$; $p<0.01$ |  |  |  |  |  |  |  |
| Resilience (Model 2) | (Constant) | 29.490 | 1.637 |  | 18.018 | 0.000 | [26.274, 32.706] |
|  | Physical Activity | 0.000 | 0.000 | 0.138 | 3.350 | 0.001 | [.00, .01] |
|  | Reappraisal | 0.734 | 0.079 | 0.381 | 9.249 | 0.000 | [.578,.891] |
| $R = .432$; $R^2 = .187$; $F_{(1,502)} = 85.539$; $p<0.01$ |  |  |  |  |  |  |  |
| Resilience (Model 3) | (Constant) | 21.606 | 1.587 |  | 13.613 | 0.000 | [18.488, 24.725] |
|  | Physical Activity | 0.000 | 0.000 | 0.060 | 1.630 | 0.104 | [.00, .01] |
|  | Reappraisal | 0.331 | 0.078 | 0.172 | 4.260 | 0.000 | [.178, .484] |
|  | Self-efficacy | 0.252 | 0.021 | 0.488 | 11.981 | 0.000 | [.210, .294] |

$R = .607$; $R^2 = .368$; $F_{(1,501)} = 143.541$; $p<0.01$; *Durbin-Watson = 1.71.*

reappraisal was included in the model, the impact level of physical activity on resilience decreased but still remained significant ($R = 0.432$; $R^2 = 0.187$; $F_{(1,502)} = 85.539$; $p = 0.000$; Effect size($f^2$) = 0.23). In Model 3, self-efficacy was finally included in the model, and it was understood that this model accounted for 36.8% of the total variance in resilience ($R = 0.611$; $R^2 = 0.373$; $F_{(1,503)} = 145.585$; $p = 0.000$; Effect size($f^2$) = 0.59) and significantly predicted resilience. The impact level of self-efficacy emerged to be higher than the impact level of the other variables ($\beta = 0.488$, $p < 0.01$). Additionally, when self-efficacy was included in the model, the impact of physical activity on resilience disappeared ($\beta = 0.060$, $p > 0.05$).

To better understand the relationships between the variables that create significant differences in psychological resilience within the models and their interactions, Jeremy Dawson's slopes were used. Jeremy Dawson's slopes are a technique used to evaluate the impact of variables in multivariate regression models. This method allows examining the contribution of each independent variable in the regression model while the other independent variables are constant. As evident from the graphs below, the most influential factor in resilience is self-efficacy.

In the slopes showing the relationships between self-efficacy, physical activity, resilience, and reappraisal, it is evident that self-efficacy creates a noticeable difference between low and high levels. For instance, as observed from Graph 1 and Graph 2, a significant difference was noticed between low and high scores of self-efficacy ($p < 0.05$), while low and high levels of physical activity and reappraisal showed similar changes. In other words, individuals with both low and high levels of physical activity and reappraisal who also exhibited high self-efficacy experienced increased resilience. The levels of physical activity and reappraisal did not create a significant difference in resilience ($p < 0.05$). Indeed, Graph 3 supports this observation. It appears that physical activity and reappraisal are not as influential on resilience as self-efficacy. In other words, self-efficacy has a serious impact on endurance compared to physical activity and reevaluation.

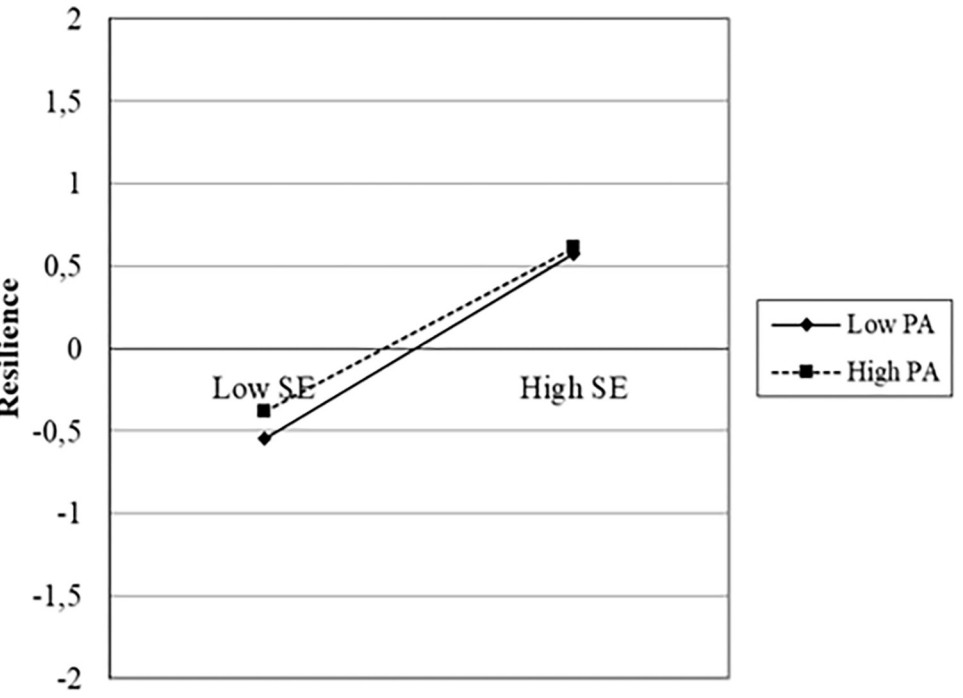

**Graph 1. Self efficacy, physical activity and resilience.**

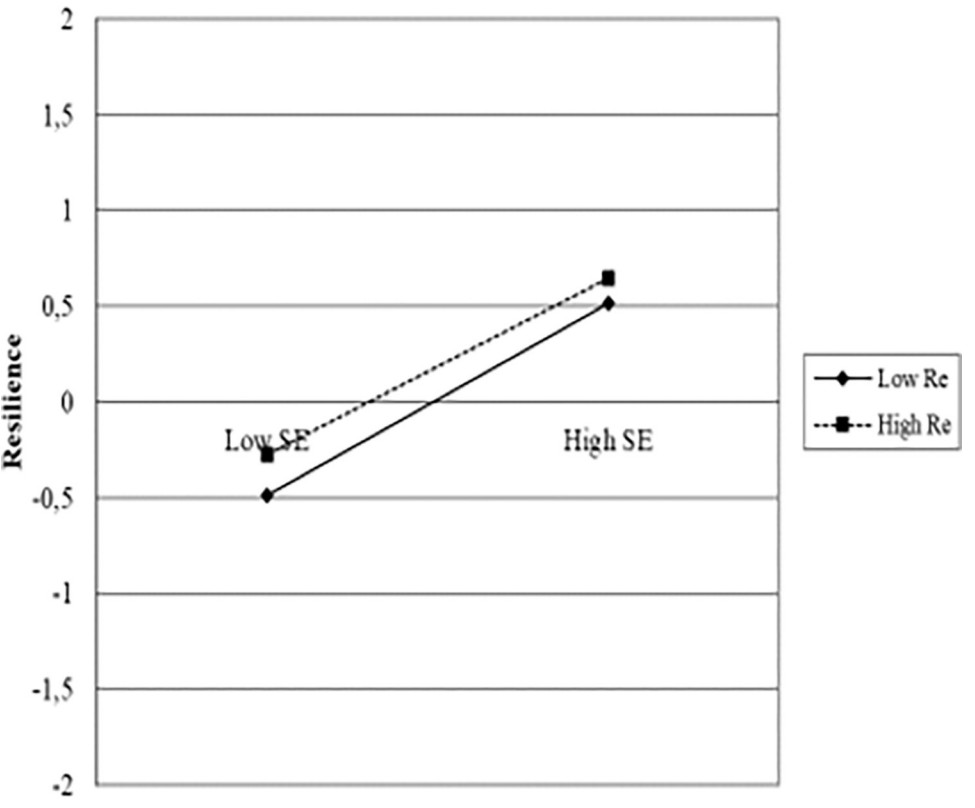

**Graph 2. Self efficacy, reappraisal and resilience.**

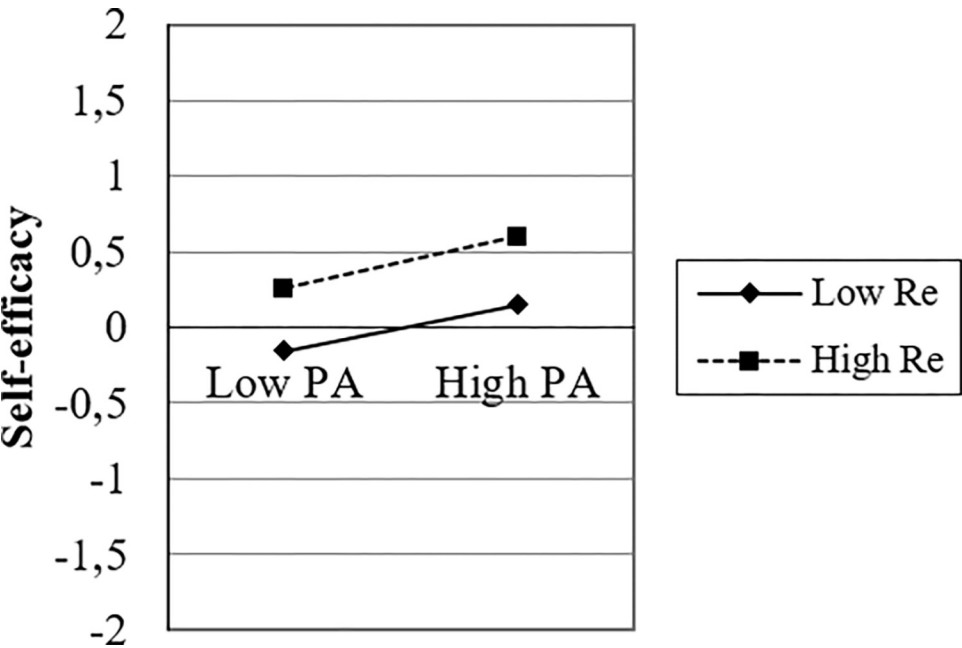

**Graph 3. Self efficacy, physical activity and reappraisal.** Not: PA = Physical Activity; SE: Self-efficacy; Re: Reappraisal.

## Discussion

During adolescence, individuals are prone to various mental and physical developmental vulnerabilities, which can be seen as constraints to maintaining a healthy lifestyle. One of the factors that promotes the physical and mental health of adolescents is psychological resilience. Through resilience, individuals can cope with negative emotions and adapt to changing external stressors throughout their lives. Therefore, interventions aimed at psychological resilience for adolescents can alleviate stress, enhance their well-being, and improve their mental health [42]. By addressing mechanisms that enhance resilience and examining the relationships between these mechanisms, the development of a healthier generation can be promoted in every aspect. This research was conducted to identify mechanisms that enhance resilience in adolescents and to determine which variables better predict resilience.

The results of the study revealed that physical activity, emotion regulation through reappraisal, and self-efficacy significantly predicted resilience. Physical activity, emotional regulation and self-efficacy are important factors in the development of resilience during adolescence. This finding aligns with the idea that promoting physical activities that enhance resilience can improve the mental health of adolescents [27]. Previous research has shown that frequent physical activity improves adolescents' resilience, which is consistent with the current study's findings [27,54]. For example, a study of 1100 Norwegian adolescents found that resilience is associated with the frequency of physical activity through structured style (planning, structure, and daily routines) in male participants [55]. According to Phillips et al. [56], physical activity is an important domain for resilience. A more recent longitudinal study found that, while there was no significant gender difference, physical activity was associated with and had a significant predictive effect on adolescents' resilience [9].

It is crucial for adolescents to develop and utilize positive self-regulation strategies to enhance their resilience [33]. Indeed, according to the research results, adolescents' emotional regulation significantly predicted their resilience through reappraisal. This result suggests that adolescents who can think in a different way when feeling bad or happy may be more psychologically resilient. Earlier research has identified a remarkable number of factors that contribute to adolescent resilience, and reappraisal is among these factors. For instance, Kuhlman et al. [57] mentioned cognitive reappraisal as a predictor of resilience during Covid-19. In their study conducted with 164 adolescents aged between 13–16 Mestre et al. [31] demonstrated that emotion regulation ability and cognitive regulation strategies, such as positive reappraisal, predicted perceived resilience among participants. Resilience and reappraisal have a reciprocal relationship and also have an impact on distinct concepts when considered collectively. For instance, in a current study it was found that coping style affected mental health through the independent mediating effects of cognitive reappraisal and psychological resilience [58]. Another study results indicated that resilience was correlated positively with cognitive reappraisal and negatively with expressive suppression and self-esteem partially mediated this relationship [59].

According to the analyzed results, adolescents' self-efficacy influences their resilience. This result can be interpreted as indicating that adolescents with higher beliefs in their academic, social, and emotional capacities are more resilient. According to Schwarzer and Warner [39], self-efficacy influences how people feel, think, and behave. Individuals with high levels of perceived self-efficacy believe in their own abilities to overcome adversity and view problems as challenges rather than threats or uncontrollable situations. Present study result is consistent with that of Sagone et al. [40], who emphasized the importance of self-efficacy in life skills like coping with challenges. However, some studies in the literature show slightly different results.

For instance, in her study Nowicki [60] discovered a positive relationship between resilience and self-efficacy in 60 adolescents, but self-efficacy did not predict resilience.

## Conclusion

The present demonstrated the effects of physical activity, emotion regulation, and self-efficacy on resilience, in line with the literature. What distinguishes this study from the existing literature is the varying levels of impact of the variables on resilience and how the inclusion of variables in sequence changes the effectiveness of one another. Both the analyses and the drawn slopes demonstrated that self-efficacy was more effective for psychological resilience than physical activity and emotion regulation. Although physical activity and emotion regulation strategies also had impacts on resilience, these impacts were not as strong as self-efficacy. Additionally, when self-efficacy was included in the model (Model 3), the impact of physical activity on resilience disappeared, indicating that physical activity alone cannot be a predictor without self-efficacy. Of course, we cannot disregard the impact of physical activity on resilience; this has already been demonstrated in both the existing literature and the findings of this study. However, without self-efficacy, no change can be made in psychological resilience, no matter how much physical activity is included. Graph 1 shows that low and high levels of physical activity show similar changes in psychological resilience, but there is a noticeable difference between low and high self-efficacy. The same applies to emotion regulation. Although the impact of emotion regulation on resilience continued in Model 3, as understood from Graph 2, the presence of low or high reappraisal did not cause any change in psychological resilience compared to low or high self-efficacy.

Self-efficacy, emotion regulation, physical activity, and resilience are all linked in a detailed and organized way in studies which analyzed these relationships. When these studies are investigated with this manner, it's easier to understand what we've found [61–63]. Physical activity, emotional regulation, and self-efficacy are all important factors in the development of resilience during adolescence. According to studies, physical activity reduces stress hormones and increases resilience, making stress management easier [64]. Furthermore, controlled emotions help adolescents cope with difficulties and make better decisions [65]. Finally, self-efficacy plays an active role in the development of resilience by positively influencing processes such as self-confidence and motivation to succeed [66]. So, it can be said that, physical activity, emotional regulation, and self-efficacy work together to improve adolescents' resilience. Providing a solid basis in these domains helps adolescents cope with challenges and live healthier lives.

## Limitations and future studies

Although the present study revealed unique interactions between resilience, physical activity, emotion regulation and self-efficacy, has some limitations. Although data were collected from 16 different provinces across the country, convenience sampling has its limitations. Limitations related to the generalisability of the data can be reduced by using probability sampling methods. The cross-sectional nature of this study means that the data was obtained within a specific time frame. So it does not Therefore, there is a need for more comprehensive and longitudinal studies examining the determinants of psychological resilience. Finally, despite excluding incomplete and erroneous data from the analysis, the subjective nature of the physical activity questionnaire may lead to bias. This limitation can be considered an important constraint regarding the generalizability of the data. In future research, the model discussed in this study can be investigated in the sample of children. Again, the mediating roles of emotion regulation and self-efficacy in the relationship between physical activity and resilience can be tested.

## Practical implications

Physical exercise is essential for a healthy body. As a matter of fact, research findings also support this view. Only 60 minutes of exercise per week can reduce depression by 12 per cent [67]. Therefore, adolescents are recommended to exercise regularly so that their resilience can improve. Adolescents are recommended to receive training in emotion regulation skills that increase positive emotions and decrease negative emotions, so that their resilience can improve. Finally, adolescents are recommended to participate in activities and events that will improve their self-efficacy.

## Author Contributions

**Conceptualization:** Gokhan Cakir, Utku Isik, Umit Dogan Ustun, Nihan Su, Osman Gumusgul.

**Data curation:** Gokhan Cakir, Utku Isik, Umit Dogan Ustun, Nihan Su.

**Formal analysis:** Gokhan Cakir, Utku Isik, Nihan Su.

**Investigation:** Nihan Su.

**Methodology:** Gokhan Cakir, Utku Isik, Umit Dogan Ustun, Nihan Su.

**Resources:** Umit Dogan Ustun, Osman Gumusgul.

**Software:** Utku Isik.

**Supervision:** Gokhan Cakir.

**Visualization:** Utku Isik.

**Writing – original draft:** Gokhan Cakir, Utku Isik.

**Writing – review & editing:** Umit Dogan Ustun, Nihan Su, Osman Gumusgul.

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
