## [Decision Letter · Decision Letter 0]

15 Mar 2024

PONE-D-24-07151Resilliance among Turkish adolescents: a multi-level approachPLOS ONE

Dear Dr. ISIK,

Thank you for submitting your manuscript to PLOS ONE. After careful consideration, we feel that it has merit but does not fully meet PLOS ONE’s publication criteria as it currently stands. Therefore, we invite you to submit a revised version of the manuscript that addresses the points raised during the review process.

We look forward to receiving your revised manuscript.

Kind regards,

Roghieh Nooripour, Ph.D

Academic Editor

PLOS ONE

Journal Requirements:

2. In the Methods ethics statement, you specified that verbal consent was obtained. Please provide additional details regarding how this consent was documented and witnessed, and state whether the IRB approved this.

Reviewers' comments:

Reviewer's Responses to Questions

**Comments to the Author**

1. Is the manuscript technically sound, and do the data support the conclusions?

Reviewer #1: Yes

Reviewer #2: Yes

2. Has the statistical analysis been performed appropriately and rigorously? 

Reviewer #1: Yes

Reviewer #2: Yes

3. Have the authors made all data underlying the findings in their manuscript fully available?

Reviewer #1: Yes

Reviewer #2: Yes

4. Is the manuscript presented in an intelligible fashion and written in standard English?

Reviewer #1: Yes

Reviewer #2: Yes

5. Review Comments to the Author

Reviewer #1: I have comprehensive review of your paper. I have provided constructive feedback aimed at enhancing the quality and impact of your article. I encourage you to consider these suggestions to improve clarity and overall effectiveness. Upon implementing revisions, I am eager to review the updated version. Your commitment to refining your work is admirable, and I am eager to witness the evolution of your article. Best regards.

Abstract

1. Provide more specific information about the findings. For example, instead of stating that physical exercise, emotion regulation, and self-efficacy are significant indicators of resilience, provide some numerical data or effect sizes to demonstrate the strength of these relationships.

INTRODUCTION

• Some statements are quite broad and could benefit from more specific details or examples. For instance, when discussing the impact of emotional regulation on resilience, you mention "unreasonable and harmful behaviors." Providing specific examples or studies to illustrate this point could strengthen your argument.

• To enhance the quality of your introduction, consider incorporating the following references

1. https://link.springer.com/article/10.1186/s40359-023-01379-w

2. https://link.springer.com/article/10.1007/s10943-020-01151-z

3. https://brieflands.com/articles/ijhrba-93481

4. https://bmcpsychology.biomedcentral.com/articles/10.1186/s40359-022-00852-2

5. https://link.springer.com/article/10.1007/s12646-023-00713-x

• The hypotheses formulated in the theoretical background section are relevant and logically derived from the literature. However, consider providing a brief justification or rationale for each hypothesis to clarify the expected relationships between variables.

• Consider adding a brief concluding paragraph to summarize the main points discussed in the introduction and reiterate the significance of the study. This will provide closure to the section and reinforce the relevance of the research objectives.

Method

1. Clarity and Organization: The section is well-organized and provides detailed information about each aspect of the study. However, consider breaking down lengthy paragraphs into smaller, more digestible chunks to improve readability. Bullet points or subheadings can be used to delineate different aspects of the methodology, making it easier for readers to follow the process.

2. The use of G*Power to determine the required sample size is appropriate and adds to the rigor of the study. However, it would be beneficial to provide a brief rationale for why a sample size of 439 participants was deemed sufficient for the study. Mentioning the expected effect size, significance level, and statistical power can help justify the chosen sample size.

3. The section on data collection procedures is clear and provides sufficient detail about the sampling method and ethical considerations. Consider briefly discussing any measures taken to ensure data quality and participant confidentiality, as these aspects are crucial in maintaining the integrity of the study.

4. The description of each data collection tool is thorough and includes information about their development and validation. However, consider providing a brief rationale for why these specific scales were chosen for the study. Discussing how each tool aligns with the research objectives and measures the constructs of interest can enhance the justification for their selection.

5. The description of data analysis procedures is detailed and includes information about the statistical tests used to analyze the data. Consider providing a rationale for why hierarchical regression analysis was chosen as the primary method for examining the relationship between independent and dependent variables. Additionally, mention any steps taken to address potential biases or confounding variables during data analysis.

6. It's commendable that assumptions for regression analysis were checked and met. However, briefly mentioning the steps taken to ensure the validity of the assumptions, such as normality, linearity, and multicollinearity, would enhance transparency and credibility.

results

• Ensure precision in reporting statistical results, including regression coefficients, significance levels, and effect sizes. Clearly state the statistical tests used and provide appropriate degrees of freedom and p-values to facilitate accurate interpretation of the findings

• Ensure consistency in the presentation of results across tables and graphs. For instance, use consistent labeling and formatting conventions to facilitate understanding and comparison of findings. Additionally, consider providing titles and legends for the graphs to clarify the variables being depicted.

• The use of Jeremy Dawson's slopes to visualize the relationships between variables is informative. Consider providing a brief explanation of how the slopes were calculated and interpreted, particularly for readers who may not be familiar with this method. Additionally, discuss the implications of the observed slopes for understanding the relative influence of different variables on resilience

Discussion

1. Consider providing more context for the findings by discussing them in relation to the broader literature on adolescent resilience, physical activity, emotion regulation, and self-efficacy. Highlight how the current study contributes to existing knowledge and what new insights it offers to the field.

2. Provide deeper interpretation of the results by discussing the underlying mechanisms that may explain the observed relationships between variables. For example, discuss why physical activity, emotion regulation, and self-efficacy are important predictors of resilience during adolescence, drawing on theoretical frameworks and empirical evidence.

3. Discuss the practical implications of the findings for interventions aimed at promoting resilience among adolescents. Consider how the insights gained from the study can inform the design and implementation of programs and policies targeting physical activity promotion, emotion regulation skills training, and self-efficacy enhancement in adolescent populations.

4. Provide a thorough discussion of the study limitations and suggest directions for future research to address these limitations. Consider discussing potential methodological improvements, such as using longitudinal designs, implementing more rigorous sampling methods, or employing objective measures of physical activity.

5. Acknowledge the limitations of generalizability inherent in convenience sampling and discuss how these limitations may impact the interpretation and application of the study findings. Consider discussing the potential differences between the study sample and the broader adolescent population and how these differences may affect the generalizability of the results.

6. Conclude the discussion by summarizing the key findings, reiterating their significance, and emphasizing the implications for theory, practice, and future research. Provide a clear and concise summary of the main takeaways from the study and how they contribute to advancing knowledge in the field of adolescent resilience.

Reviewer #2: The paper covers an interesting and important topic - identifying whether three factors: physical exercise,

emotion control, or self-efficacy impact on resilience. The age of the sample, 14-19 years (average 15.66 years) represents a crucial period in development in terms of emotional and behavioural problems; both can impact negatively on later mental health. Resilience is viewed as a protective factor against adverse mental health outcome. It also impacts on social development. More resilience indicates, for example, ability to take responsibility. The analysis showed self-efficacy as having a greater impact on resilience for the sample than did the other two variables.

The authors covered the background well with appropriate literature cited. The analysis, using correlations and linear regression, were appropriate. The results were presented clearly with tables and figures summarising the findings. As the authors state, resilience develops. The findings from the study indicate the need to assist children and adolescents develop ways of coping with challenges, and thus increase their resilience.

The authors point out the subjective nature of the physical activity questions. I have no suggestions for change, but the percentage of the sample involved in physical activities was not high. In future research it might be valuable to attempt to recruit more, and in the analysis separate physical activities into team sport activities versus individual activities because of the different social contexts. A possible question for the future: ‘Would those involved in team sports be more likely to show resilience because they are working together as a team?’

6. PLOS authors have the option to publish the peer review history of their article (what does this mean?). If published, this will include your full peer review and any attached files.

Reviewer #1: No

Reviewer #2: No

---

## [Author Response · Author response to Decision Letter 0]

27 Apr 2024

Thank you very much for your meticulous review. This review helped us move our manuscript to a better position. All requested changes and responses to changes have been added to the attachments section.

Best wishes.

---

## [Decision Letter · Decision Letter 1]

9 Jun 2024

PONE-D-24-07151R1Resilliance among Turkish adolescents: a multi-level approachPLOS ONE

Dear Dr. Isık,

Thank you for submitting your manuscript to PLOS ONE. After careful consideration, we feel that it has merit but does not fully meet PLOS ONE’s publication criteria as it currently stands. Therefore, we invite you to submit a revised version of the manuscript that addresses the points raised during the review process.

We look forward to receiving your revised manuscript.

Kind regards,

Abdullah Sarman, Assistant Professor, RN, Ph.D.

Academic Editor

PLOS ONE

Journal Requirements:

Reviewers' comments:

Reviewer's Responses to Questions

**Comments to the Author**

1. If the authors have adequately addressed your comments raised in a previous round of review and you feel that this manuscript is now acceptable for publication, you may indicate that here to bypass the “Comments to the Author” section, enter your conflict of interest statement in the “Confidential to Editor” section, and submit your "Accept" recommendation.

Reviewer #1: All comments have been addressed

Reviewer #3: All comments have been addressed

2. Is the manuscript technically sound, and do the data support the conclusions?

Reviewer #1: Yes

Reviewer #3: Yes

3. Has the statistical analysis been performed appropriately and rigorously? 

Reviewer #1: Yes

Reviewer #3: Yes

4. Have the authors made all data underlying the findings in their manuscript fully available?

Reviewer #1: Yes

Reviewer #3: Yes

5. Is the manuscript presented in an intelligible fashion and written in standard English?

Reviewer #1: Yes

Reviewer #3: Yes

6. Review Comments to the Author

Reviewer #1: The requested corrections have been completed and tested.

The article is now acceptable.

Thank you for your time.

Reviewer #3: My suggestions for the article;

Statistical values (numbers) should be removed from the abstract

The theoretical structure of resilience is clearly emphasized in the introduction.

How was sampling decided in the method? Must be explained, power analysis can be done

The inclusion and exclusion criteria for the study should be clarified

The name of the institution should be blinded in the ethics section

Articles and tables should generally be reviewed according to the journal writing rules.

The presentation of Table 2 is not appropriate, means and SDs should be shown in different places in Table 2.

Confidence interval values should be added to Table 3, Durbin Watson values of the models should be presented.

The discussion is meticulous and blended with current sources.

7. PLOS authors have the option to publish the peer review history of their article (what does this mean?). If published, this will include your full peer review and any attached files.

Reviewer #1: No

Reviewer #3: No

---

## [Author Response · Author response to Decision Letter 1]

10 Jun 2024

We thank the referees for their valuable comments. After the revised, our manuscript became much higher quality. We made the necessary revised and added the necessary documents. We look forward to your positive feedback.

---

## [Decision Letter · Decision Letter 2]

13 Jun 2024

Resilliance among Turkish adolescents: a multi-level approach

PONE-D-24-07151R2

Dear Dr. Isık,

We’re pleased to inform you that your manuscript has been judged scientifically suitable for publication and will be formally accepted for publication once it meets all outstanding technical requirements.

Kind regards,

Abdullah Sarman, Assistant Professor, RN, Ph.D.

Academic Editor

PLOS ONE

Additional Editor Comments (optional):

Reviewers' comments:

Reviewer's Responses to Questions

**Comments to the Author**

1. If the authors have adequately addressed your comments raised in a previous round of review and you feel that this manuscript is now acceptable for publication, you may indicate that here to bypass the “Comments to the Author” section, enter your conflict of interest statement in the “Confidential to Editor” section, and submit your "Accept" recommendation.

Reviewer #3: All comments have been addressed

2. Is the manuscript technically sound, and do the data support the conclusions?

Reviewer #3: Yes

3. Has the statistical analysis been performed appropriately and rigorously? 

Reviewer #3: Yes

4. Have the authors made all data underlying the findings in their manuscript fully available?

Reviewer #3: Yes

5. Is the manuscript presented in an intelligible fashion and written in standard English?

Reviewer #3: Yes

6. Review Comments to the Author

Reviewer #3: Dear author

Thank you for carefully studying the revision suggestions.

I have no additional suggestions for your article.

7. PLOS authors have the option to publish the peer review history of their article (what does this mean?). If published, this will include your full peer review and any attached files.

Reviewer #3: No

---

## [Editor Report · Acceptance letter]

19 Jun 2024

PONE-D-24-07151R2 

PLOS ONE

Dear Dr. ISIK, 

I'm pleased to inform you that your manuscript has been deemed suitable for publication in PLOS ONE. Congratulations! Your manuscript is now being handed over to our production team.

Kind regards, 

on behalf of

Dr. Abdullah Sarman 

Academic Editor

PLOS ONE